

# Integrating morphological and molecular approaches for characterizing four species of *Dactylogyrus* (Monogenea: Dactylogyridae) from Moroccan cyprinids, with comments on their host specificity and phylogenetic relationships

Eva Řehulková[1], Imane Rahmouni[2], Antoine Pariselle[3,4] and Andrea Šimková[1]

[1] Department of Botany and Zoology, Faculty of Science, Masaryk University, Brno, Czech Republic
[2] Laboratory of Biodiversity, Ecology and Genome, Faculty of Sciences, Mohammed V University, Rabat, Morocco
[3] Institute of Evolutionary Sciences of Montpellier (ISEM), University of Montpellier, CNRS, IRD, Montpellier, France
[4] Faculty of Sciences, Mohammed V University, Rabat, Morocco

Corresponding author
Eva Řehulková, evar@sci.muni.cz

## ABSTRACT

Cyprinid fishes are known to harbour highly host-specific gill-associated parasites of *Dactylogyrus*. High similarity in the morphology of sclerotized structures among *Dactylogyrus* species, especially those parasitizing congeneric cyprinoids, makes their identification difficult. In this paper, four previously known species of *Dactylogyrus* are characterized and illustrated under a reliable taxonomic framework integrating morphological and molecular evidence, and their phylogenetic relationships are investigated using molecular data. The species are as follows: *D. borjensis* from *Luciobarbus zayanensis*; *D. draaensis* from *Luciobarbus lepineyi*; *D. ksibii* from *Luciobarbus ksibi* and *Luciobarbus rabatensis*; and *D. marocanus* from *Carasobarbus fritschii, L. ksibi, L. zayanensis* and *Pterocapoeta maroccana*. Our results revealed intraspecific genetic variability among specimens of *D. ksibii* collected from two different hosts and geographically distant basins. Phylogenetic reconstruction showed that *Dactylogyrus* spp. parasitizing Moroccan cyprinids are representatives of three main lineages corresponding to morphological differences and host specificity. Our records of *D. marocanus* on *L. zayanensis* and *P. maroccana* increase the range of available host species i.e.,eight species of four cyprinid genera representing two phylogenetic lineages (i.e., Barbinae and Torinae).

## INTRODUCTION

Species of *Dactylogyrus* Diesing, 1850 are ectoparasitic flatworms (Monogenea, Platyhelminthes) occurring mainly on the gills of cyprinid fishes. With more than 900

nominal species, *Dactylogyrus* represents the largest helminth genus (*Gibson, Timofeeva & Gerasev, 1996*). Recent studies on northwest African cyprinids have shown that the biodiversity of this fish group is higher than previously estimated (e.g., *Brahimi et al., 2018*; *Casal-López et al., 2015*; *Doadrio et al., 2016*; *Doadrio, Casal-López & Perea, 2016*). In Morocco, 20 species belonging to four genera (i.e., *Carasobarbus* Karaman, 1971; *Labeobarbus* Rüppel, 1835; *Luciobarbus* Heckel, 1843 and *Pterocapoeta* Günther, 1902) are currently considered valid (*Fricke, Eschmeyer & Van der Laan, 2020*). Among these, *Luciobarbus* is the most diverse, with 15 species, some being only recently described (*Brahimi et al., 2018*; *Casal-López et al., 2015*; *Doadrio et al., 2016*; *Doadrio, Casal-López & Perea, 2016*). Descriptions of the new species may indicate that the diversity of their host-specific parasites, such as monogeneans of *Dactylogyrus*, is currently underestimated. To date, 17 *Dactylogyrus* spp. parasitizing 16 cyprinid species of three genera (*Carasobarbus, Labeobarbus* and *Luciobarbus*) have been recorded in Morocco. Of these, 11 species are restricted to a single host species, five occur on two to six species belonging to one host genus and one species parasitizes species of three host genera (*El Gharbi, Birgi & Lambert, 1994*; *Rahmouni et al., 2017*).

In 2015, a survey was initiated to determine the diversity of *Dactylogyrus* species parasitizing Moroccan cyprinids. A total of 13 cyprinid species were examined and 13 species of *Dactylogyrus* were collected and prepared for morphological and molecular analysis. The first paper stemming from this investigation included descriptions of four new *Dactylogyrus* species infesting three species of northern Moroccan *Luciobarbus* (*Rahmouni et al., 2017*). This study also demonstrated that an integrated morphological and molecular approach to species identification/description can reveal the presence of morphologically indistinguishable, but genetically distinct, species (or cryptic species) within *Dactylogyrus*. *Rahmouni et al. (2017)* reported two cryptic species (*Dactylogyrus benhoussai* and *Dactylogyrus varius* forma vulgaris) parasitizing allopatric species of *Luciobarbus* (*L. yahyaouii* [syn. *L. moulouyensis*] and *L. maghrebensis*, respectively). On the other hand, they also identified three morphologically distinct, but genetically identical, forms of *D. varius*, demonstrating the usefulness of molecular markers for documenting levels of intraspecific morphological variability or, in extreme cases, to demonstrate that two or more monogenean morphospecies (morphotypes) represent a single species.

More recently, *Šimková et al. (2017)* used 10 species of *Dactylogyrus* parasitizing Moroccan cyprinids to infer potential historical contacts between northwest African, European and Asian cyprinid faunas. Using phylogenetic reconstruction, they suggested that *Dactylogyrus* species infecting Moroccan species of *Carasobarbus* were phylogenetically closely related to some *Dactylogyrus* spp. parasitizing Iberian *Luciobarbus* spp. and shared a common ancestor with southern and southeast Asian Labeonini hosts, while *Dactylogyrus* species infecting Moroccan *Luciobarbus* spp. formed a large group together with *Dactylogyrus* species parasitizing European *Luciobarbus* (including some Iberian and two known Balkan species), *Barbus* and leuciscid hosts.

The present paper is a continuation of our research dealing with *Dactylogyrus* species from Moroccan cyprinids. Herein, an effort is made to characterize and illustrate four previously known species of *Dactylogyrus* under a reliable taxonomic framework integrating

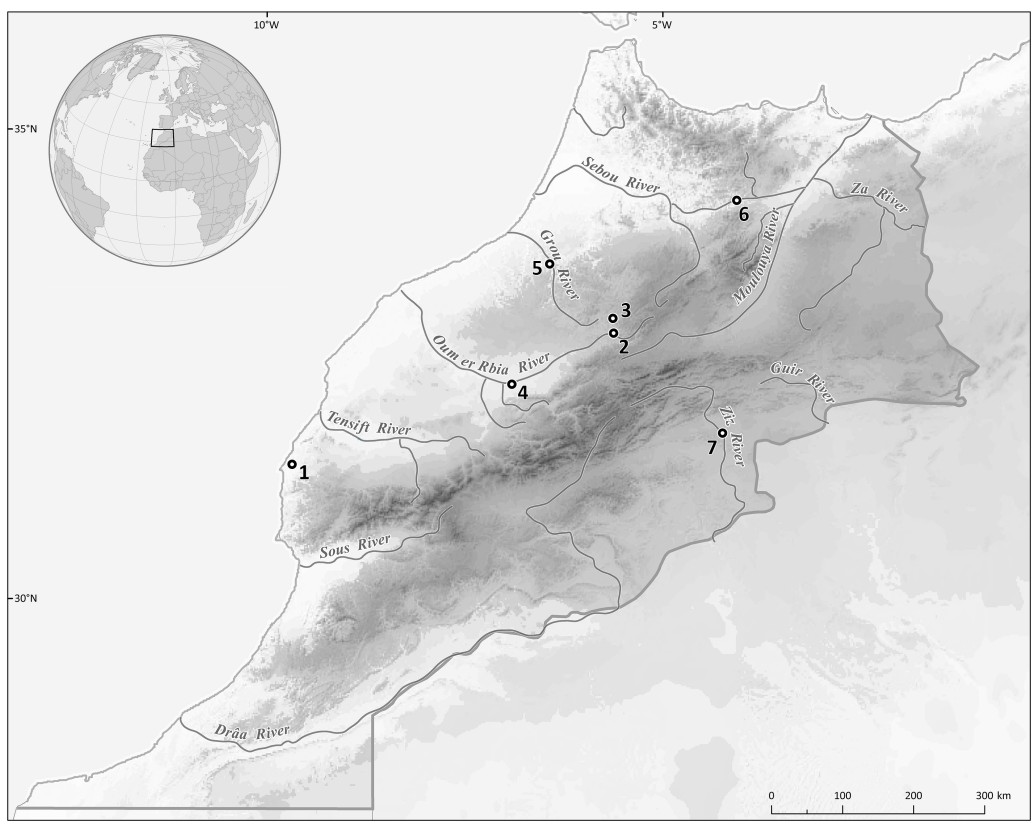

**Figure 1** Map showing the sampling localities: 1, Ksob River; 2, Chbouka River; 3, Oum Er'Rabia (near El Borj); 4, Oum Er'Rabia (Dar Oulad Zidouh); 5, Grou River; 6, Lahdar River; 7, Zouala Oasis.

morphological and molecular evidence and their phylogenetic relationships investigated using molecular data.

## MATERIALS & METHODS

### Fish sampling

Two hundred specimens of six cyprinid species, namely *Carasobarbus fritschii* (Günther, 1874); *Luciobarbus ksibi* (Boulenger, 1905); *Luciobarbus lepineyi* (Pellegrin, 1939); *Luciobarbus rabatensis* Doadrio, Perea & Yahyaoui, 2015; *Luciobarbus zayanensis* Doadrio Casal-López & Yahyaoui, 2016; and *Pterocapoeta maroccana* Günther, 1902, were captured by means of gill nets or electro-fishing from seven localities in Morocco (Fig. 1, Table 1). The scientific names and classification of fishes used are those provided in *Fricke, Eschmeyer & Van der Laan (2020)*; names used in the original description are retained in square brackets as synonyms. Live fishes were kept in aerated holding tanks until processed for parasitological examination. The research was approved by the Ethics Committee of Masaryk University (approval number CZ01302). Field experiments were approved by the Haut Commissairiat aux Eaux et Forêts et à la Lutte contre la Désertification (Ministère

**Table 1  List of hosts and their sampling localities.**

| Host species | Locality name and number | River Basin | Coordinates |
| --- | --- | --- | --- |
| *Carasobarbus fritschii* | Ksob River (**1**) | Ksob | 31°27′50.7″N, 9°45′25.3″W |
| | Chbouka River (**2**) | Oum Er'Rabia | 32°51′32.9″N, 5°37′18.9″W |
| | Lahdar River (**6**) | Sebou | 34°15′30.1″N, 4°03′52.1″W |
| *Luciobarbus ksibii* | Ksob River (**1**) | Ksob | 31°27′50.7″N, 9°45′25.3″W |
| | Chbouka River (**2**) | Oum Er'Rabia | 32°51′32.9″N, 5°37′18.9″W |
| *Luciobarbus lepineyi* | Zouala Oasis (**7**) | Ziz, Guir and Ghéris | 31°47′31.9″N, 4°14′43.5″W |
| *Luciobarbus rabatensis* | Grou River (**5**) | Bouregreg | 33°35′28.1″N, 6°25′43.7″W |
| *Luciobarbus zayanensis* | Oum Er'Rabia River (El Borj) (**3**) | Oum Er'Rabia | 33°00′58.7″N, 5°37′48.6″W |
| | Oum Er'Rabia River (Dar Oulad Zidouh) (**4**) | Oum Er'Rabia | 32°18′54.0″N, 6°54′28.7″W |
| *Pterocapoeta maroccana* | Oum Er'Rabia River (El Borj) (**3**) | Oum Er'Rabia | 33°00′58.7″N, 5°37′48.6″W |

de l'Agriculture, de la Pêche Maritime, du Développement Rural et des Eaux et Forêts, Royaume du Maroc) (N°62 HCEFLCD/DLCDPN/CPC/PPC).

## Parasite collection and fixation

Fishes were sacrificed by severing the spinal cord, after which the gill arches were removed via dorsal and ventral section and examined for monogeneans using a stereomicroscope. Specimens of *Dactylogyrus* were detached from the gills using fine needles and prepared following *Řehulková (2018)*. Monogeneans, fixed with a mixture of glycerine and ammonium picrate (GAP) (*Malmberg, 1957*), were observed under an Olympus BX51 microscope equipped with phase contrast optics. Specimens were measured using ImageJ software (available at: http://rsb.info.nih.gov/ij/) following *Řehulková, Benovics & Šimková (2020)*. Measurements, all in micrometers, are expressed as the mean followed in parentheses by the range and number (*n*) of structures measured. Numbering of hook pairs was adopted from *Mizelle (1936)*. The male copulatory organ is henceforth abbreviated to MCO. Voucher specimens of monogeneans collected in the present study were deposited at the Muséum National d'Histoire Naturelle (MNHN), Paris, France and the Institute of Parasitology of the Czech Academy of Sciences (IPCAS), České Budějovice, Czech Republic as indicated in the respective species accounts.

## DNA isolation, amplification and sequencing

To guarantee identification of parasites collected for molecular analysis, specimens collected for DNA extraction were bisected into two parts, with one part transferred to an Eppendorf tube containing ethanol (96%) to preserve the DNA and the second half mounted in GAP for species identification. Individual parasites were dried using a vacuum centrifuge. Genomic DNA was extracted using the DNEasy extraction kit (Qiagen) following the manufacturer's instructions, the extracted DNA being concentrated to a final volume of 80μl. Partial 18S rDNA and the entire ITS1 region were amplified using the primers S1 (5′-ATTCCGATAACGAACGAGACT-3′) and IR8 (5′-GCTAGCTGCGTTCTTCATCGA-3′), which anneal to the 18S and 5.8S rDNA sections, respectively (*Šimková et al., 2003*). PCR was carried out following the protocol and conditions described by *Rahmouni et al. (2017)*. Partial 28S rDNA was amplified using the following primers: forward C1

(5′-ACCCGCTGAATTTAAGCA-3′) and reverse D2 (5′-TGGTCCGTGTTTCAAGAC-3′) (*Hassouna, Michot & Bachellerie, 1984*). PCR was performed following *Rahmouni et al. (2017)*. Visualization, purification and sequencing of PCR products followed the methodology described in *Rahmouni et al. (2017)*.

## Sequence alignment and phylogenetic analysis

The DNA sequences obtained were analyzed using Sequencher software (Gene Codes Corp.). All sequences were aligned using Clustal W multiple alignments (*Thompson, Higgins & Gibson, 1994*) running in BioEdit version 7.2.5 (*Hall, 1999*). Genetic distances for each molecular marker (28S rDNA, 18S rDNA and ITS1), representing uncorrected p-distances between sequences of the different Moroccan species reported herein, were calculated using MEGA 7 (*Tamura et al., 2013*) to express interspecific or intraspecific variability. For phylogenetic analyses, all Moroccan representatives of *Dactylogyrus* parasitizing cyprinids with available sequence data (i.e., nine *Dactylogyrus* spp. from *Luciobarbus* and four species of *Dactylogyrus* from *C. fritschii*) were included in this study. The four remaining species previously reported to parasitize Moroccan cyprinids (i.e., *D. guirensis* (*El Gharbi, Birgi & Lambert, 1994*), *D. ksibioides* (*El Gharbi, Birgi & Lambert, 1994*), *D. oumiensis* (*El Gharbi, Birgi & Lambert, 1994*), and *Dactylogyrus reinii* (*El Gharbi, Birgi & Lambert, 1994*) were not included in the analyses due to missing recent records of these species and the associated absence of their DNA sequences. The majority of *Dactylogyrus* species (i.e., eleven *Dactylogyrus* species) parasitizing European species of *Barbus* and *Luciobarbus* were also included in the phylogenetic analyses. The phylogenetic analysis was performed using unambiguous alignment of combined sequences (28S rDNA, 18S rDNA and ITS1), with gaps and ambiguously aligned regions removed from the alignment using GBlocks version 0.91 (*Talavera & Castresana, 2007*). Phylogenetic analysis was performed using the maximum likelihood (ML) and Bayesian inference (BI) approaches in RAxML (*Stamatakis, 2014*) and MrBayes version 3.2.6 (*Huelsenbeck & Ronquist, 2001*), respectively. JModelTest version 2.1.10 (*Guindon & Gascuel, 2003*; *Darriba et al., 2012*) was employed to select the most appropriate model of DNA evolution using the Akaike information criterion (AIC). GTR+ I + G for 28S rDNA and ITS1, and TIM3ef + I + G for 18S rDNA, were selected as the best models of DNA evolution, using the following partition 18S = 1-446, ITS1 = $447 - 745$ and 28S = $746 - 1533$. GTR model including the proportion of invariable sites and gamma distribution were set for reconstruction of phylogenetic trees in RAxML and MrBayes. Support values for internal nodes were estimated using a bootstrap resampling procedure with 1,000 replicates (*Felsenstein, 1985*). The BI tree was constructed using four Monte Carlo Markov chains (MCMC) running under 2,000,000 generations, with sampling tree topologies every 100 generations. The first 30% of the trees were discarded as "burn-in" according to the standard deviation split frequency value ($<0.01$). The posterior probabilities of the phylogeny and its branches were determined for all trees left in the plateau phase with the best ML scores.

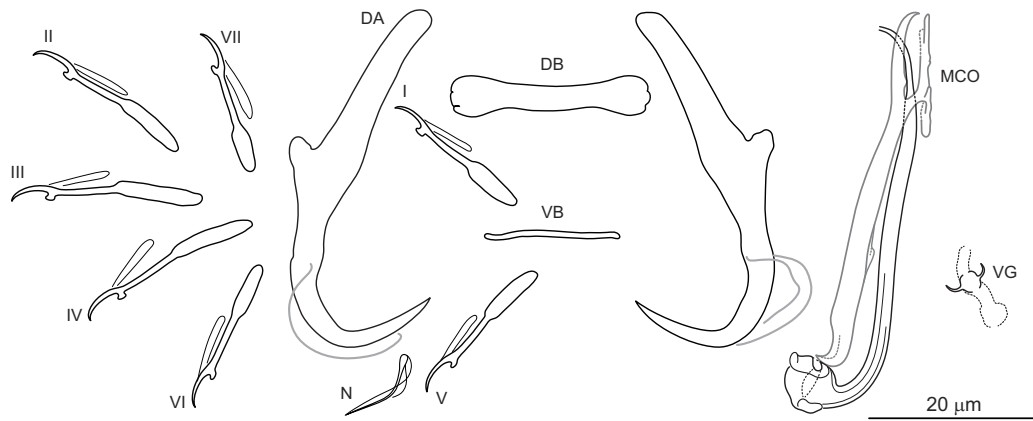

**Figure 2** **Sclerotized structures of *Dactylogyrus marocanus* ex *Pterocapoeta maroccana* (Oum Er'Rabia River, El Borj).** DA, dorsal anchor; DB, dorsal bar; VB, ventral bar; N, needle; I-VII, hooks (pairs I-V ventral; pairs VI, VII dorsal); MCO, male copulatory organ; VG, vagina.

## RESULTS

Six Moroccan cyprinid fish species were examined for monogeneans: *C. fritschii* ($n = 40$), *L. ksibi* ($n = 15$), *L. lepineyi* ($n = 113$), *L. rabatensis* ($n = 9$), *L. zayanensis* ($n = 15$) and *P. maroccana* ($n = 3$). In each species, the gills were infected with one or more species of *Dactylogyrus*. Four previously described species of *Dactylogyrus* were found (*D. marocanus*, *D. ksibii*, *D. borjensis* and *D. draaensis*) and these were taxonomically evaluated using an integrated approach combining both morphological and molecular analysis, thereby making them available for phylogenetic and coevolutionary analysis.

***Dactylogyrus marocanus*** El Gharbi, Birgi & Lambert, 1994 (Fig. 2)
**Type host:** *Carasobarbus fritschii* (Günther, 1874) [syns. *Barbus* (*Labeobarbus*) *fritschii*, *B.* (*L.*) *paytonii*].
**Type locality:** Oum Er'Rabia Basin, Boulaouane; Morocco.
**Other previously recorded hosts:** *Carasobarbus harterti* (Günther, 1901) [syn. *Barbus* (*Labeobarbus*) *harteti*], *Labeobarbus reinii* (Günther, 1874) [syn. *Barbus* (*Labeobarbus*) *reinii*], *Luciobarbus ksibi* (Boulenger, 1905) [syn. *Barbus* (*Barbus*) *ksibi*], *Luciobarbus nasus* (Günther, 1874) [syn. *Barbus* (*Barbus*) *nasus*], *Luciobarbus setivimensis* (Valenciennes, 1842) [*Barbus* (*Barbus*) *setivimensis*] (*El Gharbi, Birgi & Lambert, 1994*). Identification of *L. setivimensis* as host for *D. marocanus* is apparently erroneous (see Remarks for *D. ksibii*).
**Other previously recorded localities:** Oum Er'Rabia Basin (El Borj, Mechraa Ben Abbou); Tensift Basin, Zate River (Ait Ourir); Ksob Basin (Essaouira); Sebou Basin, Ouargha River (Ouazzane); Moulouya Basin (Mechraa Hammadi, Aklim); Morocco (*El Gharbi, Birgi & Lambert, 1994*).
**Present hosts and localities:** *Carasobarbus fritschii* (localities 1, 2, 6), *Luciobarbus ksibi* (localities 1, 2), *Luciobarbus zayanensis* Doadrio, Casal-López & Yahyaoui, 2016 (localities 3, 4), *Pterocapoeta maroccana* Günther, 1902 (locality 3).
**Site on host:** Gill lamellae.

**Specimens deposited:** Two vouchers (IPCAS M-750) and 2 hologenophores (IPCAS M-750) from *C. fritschii* (Chbouka and Ksob River), 3 vouchers (IPCAS M-750) and 1 hologenophore (IPCAS M-750) from *L. ksibi* (Chbouka River), 3 vouchers (MNHN HEL 1255, 1267) and 3 hologenophores (MNHN HEL1258, 1259; IPCAS M-750) from *L. zayanensis* (El Borj), and 6 vouchers (MNHN HEL1256, 1257) and 2 hologenophores (IPCAS M-750) from *P. maroccana* (El Borj).

**Description:** Based on 10 specimens fixed in GAP. Body length 303 (263–348; $n = 10$); greatest width 80 (73–90; $n = 10$) at level of ovary. One pair of anchors located dorsally: total length 44 (41–50; $n = 10$); length to notch 22 (19–23; $n = 10$); inner root length 18 (15–20; $n = 10$); outer root length 2 (1–3; $n = 10$); point length 10 (9–11; $n = 10$). One pair of needles located near hooks of pair V. Dorsal bar straight, bone-shaped, with slightly enlarged lateral ends, 24 (21–27; $n = 10$) long. Ventral bar reduced into thin rod-like structure, 13 (12–15; $n = 10$) long. Seven pairs of hooks; each with delicate point, protruded thumb and shank inflated along proximal half; filamentous hook (FH) loop extending to level of shank inflation. Hook lengths: pair $I = 19$ (18–20; $n = 10$); pair II $= 19$ (17–20; $n = 10$); pair III $= 22$ (20–25; $n = 10$); pair IV $= 23$ (21–26; $n = 10$); pair $V = 18$ (16–19; $n = 10$); pair VI $=19$ (16–24; $n = 10$); pair VII 18 (15–23; $n = 10$). MCO comprising basally articulated copulatory tube and accessory piece; total straight length 44 (38–57; $n = 10$). Copulatory tube J-shaped, with bulbous base and diameter sharply attenuated distally, 47 (41–61; $n = 10$) long. Accessory piece rod-shaped, distally bifurcated to form two arms (one hook-shaped). Vagina lightly sclerotized, 5 (4–7; $n = 10$) long.

**Molecular characterization:** The following DNA sequences were obtained for *D. marocanus*: partial 28S rDNA 787 bp long, partial 18S rDNA 478 bp long, ITS1 region 483 bp long, and 5.8S rDNA 11 bp long. Eight specimens from four host species (*C. fritschii*, *L. ksibi*, *L. zayanensis* and *P. maroccana*) collected from different localities were sequenced, with no intraspecific genetic variability found between different host species or between different localities. Accession numbers of *D. marocanus* collected from different host species are included in Table 2. Pairwise genetic distances between *D. marocanus* and other Moroccan *Dactylogyrus* spp. showed very high molecular divergence (Table 3).

**Remarks:** *Dactylogyrus marocanus* was originally described from the gills of six species belonging to three cyprinid genera (*Carasobarbus*, *Labeobarbus* and *Luciobarbus*) inhabiting five large river basins in Morocco (*El Gharbi, Birgi & Lambert, 1994*; see above). However, one of the host species recorded, *L. setivimensis*, was probably misidentified as the species only occurs naturally in coastal rivers in northeast Algeria (Soummam Basin) (*Fricke, Eschmeyer & Van der Laan, 2020*). *Dactylogyrus marocanus* resembles several species belonging to the 'pseudanchoratus' species-group defined in *Paperna (1979)*, in that it has anchors with a markedly elongated inner root, short outer root and a proximally swollen shaft (an attachment point for the anchor filament). It most closely resembles *Dactylogyrus longiphallus* Paperna, 1973 from African cyprinids in that it possesses a J-shaped copulatory tube and a rod-shaped accessory piece distally bifurcated to form two arms that serve as a guide for the distal part of the copulatory tube. In addition, the MCO is markedly large in relation to the size of the haptoral structures in both species (compare with Plate XV, Figs. 1–11 in *Paperna, 1979*). However, *D. marocanus* is easily

**Table 2  List of *Dactylogyrus* species used in this study with their host species, localities of sampling, specimens deposited (HEL, MNHN; M, IP-CAS), and GenBank accession numbers (\*present study).**  Numbers of localities correspond to those in Fig. 1.

| *Dactylogyrus* spp. | Host species | Locality | Specimens deposited | | GenBank ID | |
|---|---|---|---|---|---|---|
| | | | **Vouchers** | **Hologenophores** | **28S** | **18S + ITS1** |
| *D. borjensis* | *L. zayanensis* | El Borj (3) | HEL1265-1267 | HEL1268, 1269 | MN973819 | MN974257 |
| | | Dar Oulad Zidouh (4) | M-752 | | | |
| *D. draaensis* | *L. lepiney* | Zouala Oasis (7) | HEL1351, 1352 | | MN973816 | MN974258 |
| *D. ksibii* 1 | *L. ksibi* | Ksob River (1) | HEL1260 | HEL1262 | MN973812 | MN974252 |
| *D. ksibii* 2 | *L .ksibi* | Chbouka River (2) | HEL1261 | HEL1263 | MN973811 | MN974251 |
| *D. ksibii* 3 | *L. rabatensis* | Grou River (5) | M-751 | HEL1264/ M-751 | MN973817 | MN974250 |
| *D. marocanus* | *C. fritschii* | Chbouka River (2) | M-750 | M-750 | | KY629333 |
| | | Ksob River (1) | M-750 | M-750 | KY629355 | |
| | *L. ksibi* | Chbouka River (2) | M-750 | M-750 | MW218580* | MW218673* |
| | *L. zayanensis* | El Borj (3) | HEL1255, 1267 | HEL1258, 1259/ M-750 | MW218669* | MW218671* |
| | *P. maroccana* | El Borj (3) | HEL1256, 1257 | M-750 | MW218579* | MW218672* |

differentiated from other species belonging to the 'pseudanchoratus' species-group by the presence of the ventral bar. *Dactylogyrus marocanus* demonstrates a relatively low level of host specificity, having been recorded on eight host species of four genera from Morocco, including *L. zayanensis* and *P. maroccana* reported in our study. During the present survey, no morphological differences were found between specimens of this species collected from *C. fritschii, L. ksibii, L. zayanensis* and *P. maroccana*.

**Dactylogyrus ksibii** El Gharbi, Birgi & Lambert, 1994 (Figs. 3A and 3B).
**Type host:** *Luciobarbus ksibi* (Boulenger, 1905) [syn. *Barbus* (*Barbus*) *ksibi*]
**Type locality:** Ksob Basin, Essaouira; Morocco.
**Other previously recorded hosts:** *Luciobarbus magniatlantis* (Pellegrin, 1919) [syn. *Barbus* (*Barbus*) *magniatlantis*], *Luciobarbus setivimensis* (Valenciennes, 1842) [syn. *Barbus* (*Barbus*) *setivimensis*] (*El Gharbi, Birgi & Lambert, 1994*). Identification of *L. setivimensis* as host for *D. ksibii* is apparently erroneous (see Remarks).
**Other previously recorded localities:** Oum Er'Rabia Basin (Borj, Bounual); Tensift Basin, Ourika River (Ourika), Zate River (Ait Ourir); Mellah Basin (Khouribga); Bouregreg Basin, Grou River (Moulay Bouazza), Boulahmayl River (Aguelmouss); Morocco (*El Gharbi, Birgi & Lambert, 1994*).
**Present hosts and localities:** *Luciobarbus ksibi* (Boulenger, 1905) (localities 1, 2), *Luciobarbus rabatensis* Doadrio, Perea & Yahyaoui, 2015 (locality 5).
**Site on host:** Gill lamellae.
**Specimens deposited:** Three vouchers (MNHN HEL1260) and one hologenophore (MNHN HEL1262) from *L. ksibi* (Ksob River), one voucher (MNHN HEL1261) and one hologenophore (MNHN HEL1263) from *L. ksibi* (Chbouka River), and two vouchers (IPCAS M-751) and two hologenophores (MNHN HEL1264, IPCAS M-751) from *L. rabatensis* (Grou River).

**Table 3  Uncorrected pairwise genetic distances between Moroccan species of *Dactylogyrus*.**

| *Dactylogyrus* spp. | | 1 | 2 | 3 | 4 | 5 | 6 | 7 | 8 | 9 | 10 | 11 |
|---|---|---|---|---|---|---|---|---|---|---|---|---|
| **28S rDNA** | | | | | | | | | | | | |
| 1 | *D. borjensis* | | | | | | | | | | | |
| 2 | *D. atlasensis* | 0.013 | | | | | | | | | | |
| 3 | *D. benhoussai* | 0.015 | 0.018 | | | | | | | | | |
| 4 | *D. draaensis* | 0.026 | 0.031 | 0.031 | | | | | | | | |
| 5 | *D. scorpius* | 0.019 | 0.019 | 0.012 | 0.035 | | | | | | | |
| 6 | *D. falsiphallus* | 0.015 | 0.010 | 0.021 | 0.034 | 0.021 | | | | | | |
| 7 | *D. ksibii* 1 | 0.022 | 0.024 | 0.012 | 0.035 | 0.015 | 0.026 | | | | | |
| 8 | *D. ksibii* 2 | 0.022 | 0.024 | 0.012 | 0.035 | 0.015 | 0.026 | 0.003 | | | | |
| 9 | *D. ksibii* 3 | 0.022 | 0.022 | 0.009 | 0.035 | 0.013 | 0.023 | 0.005 | 0.005 | | | |
| 10 | *D. fimbriphallus* | 0.021 | 0.026 | 0.026 | 0.015 | 0.030 | 0.028 | 0.032 | 0.032 | 0.032 | | |
| 11 | *D. varius* | 0.019 | 0.022 | 0.004 | 0.035 | 0.015 | 0.024 | 0.015 | 0.015 | 0.013 | 0.030 | |
| 12 | *D. marocanus* | 0.201 | 0.204 | 0.202 | 0.208 | 0.202 | 0.204 | 0.206 | 0.208 | 0.203 | 0.201 | 0.206 |
| **ITS1** | | | | | | | | | | | | |
| 1 | *D. borjensis* | | | | | | | | | | | |
| 2 | *D. atlasensis* | 0.067 | | | | | | | | | | |
| 3 | *D. benhoussai* | 0.043 | 0.057 | | | | | | | | | |
| 4 | *D. draaensis* | 0.128 | 0.133 | 0.126 | | | | | | | | |
| 5 | *D. scorpius* | 0.062 | 0.078 | 0.043 | 0.150 | | | | | | | |
| 6 | *D. falsiphallus* | 0.036 | 0.071 | 0.050 | 0.126 | 0.064 | | | | | | |
| 7 | *D. ksibii* 1 | 0.055 | 0.074 | 0.033 | 0.140 | 0.038 | 0.057 | | | | | |
| 8 | *D. ksibii* 2 | 0.055 | 0.074 | 0.033 | 0.140 | 0.038 | 0.057 | 0.000 | | | | |
| 9 | *D. ksibii* 3 | 0.055 | 0.074 | 0.033 | 0.140 | 0.043 | 0.059 | 0.014 | 0.014 | | | |
| 10 | *D. fimbriphallus* | 0.078 | 0.086 | 0.076 | 0.112 | 0.095 | 0.093 | 0.093 | 0.093 | 0.090 | | |
| 11 | *D. varius* | 0.059 | 0.071 | 0.033 | 0.143 | 0.048 | 0.057 | 0.043 | 0.043 | 0.048 | 0.093 | |
| 12 | *D. marocanus* | 0.428 | 0.418 | 0.430 | 0.437 | 0.439 | 0.432 | 0.439 | 0.439 | 0.442 | 0.425 | 0.432 |
| **18S rDNA** | | | | | | | | | | | | |
| 1 | *D. borjensis* | | | | | | | | | | | |
| 2 | *D. atlasensis* | 0.002 | | | | | | | | | | |
| 3 | *D. benhoussai* | 0.002 | 0.004 | | | | | | | | | |
| 4 | *D. draaensis* | 0.023 | 0.025 | 0.025 | | | | | | | | |
| 5 | *D. scorpius* | 0.004 | 0.006 | 0.006 | 0.023 | | | | | | | |
| 6 | *D. falsiphallus* | 0.004 | 0.006 | 0.006 | 0.023 | 0.004 | | | | | | |
| 7 | *D. ksibii* 1 | 0.006 | 0.008 | 0.008 | 0.025 | 0.008 | 0.010 | | | | | |
| 8 | *D. ksibii* 2 | 0.006 | 0.008 | 0.008 | 0.027 | 0.010 | 0.010 | 0.000 | | | | |
| 9 | *D. ksibii* 3 | 0.006 | 0.008 | 0.008 | 0.025 | 0.008 | 0.010 | 0.002 | 0.002 | | | |
| 10 | *D. fimbriphallus* | 0.010 | 0.012 | 0.012 | 0.029 | 0.014 | 0.014 | 0.017 | 0.017 | 0.017 | | |
| 11 | *D. varius* | 0.000 | 0.002 | 0.002 | 0.023 | 0.004 | 0.004 | 0.004 | 0.004 | 0.006 | 0.010 | |
| 12 | *D. marocanus* | 0.043 | 0.041 | 0.045 | 0.066 | 0.048 | 0.043 | 0.050 | 0.050 | 0.050 | 0.054 | 0.043 |

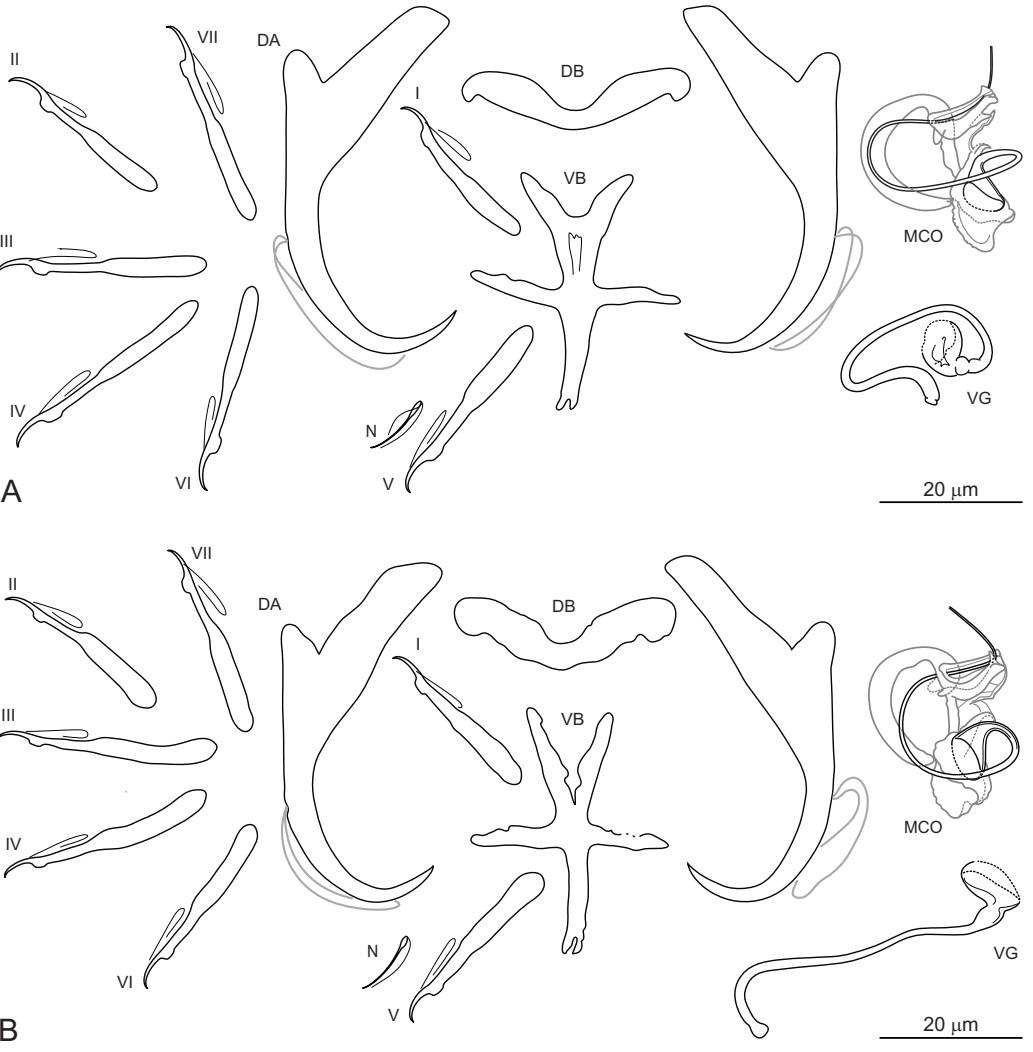

**Figure 3** **Sclerotized structures of *Dactylogyrus ksibii* ex *Luciobarbus ksibi* (Ksob River) (A) and *L. rabatensis* (Grou River) (B).** DA, dorsal anchor; DB, dorsal bar; VB, ventral bar; N, needle; I–VII, hooks (pairs I–V ventral; pairs VI, VII dorsal); MCO, male copulatory organ; VG, vagina.

**Description:** Based on 20 specimens fixed in GAP. Body length 630 (580–670; $n = 20$); greatest width 128 (120–136; $n = 20$) at level of ovary. One pair of anchors located dorsally: total length 50 (45–53; $n = 20$); length to notch 43 (39–45; $n = 20$); inner root 20 (17–22; $n = 20$) long; outer root 7 (5–8; $n = 20$) long; shaft curved; point 10 (9–11; $n = 20$) long. One pair of needles located near hook pair V. Dorsal bar broadly V-shaped, with slightly narrowed median part, 32 (29–35; $n = 20$) long. Ventral bar cross-shaped, with five arms, 34 (32–36; $n = 20$) long, 27 (24–30; $n = 20$) wide. Seven pairs of hooks; each with delicate point, flattened thumb and shank inflated along proximal two-thirds; FH loop extending to level of shank inflation. Hook lengths: pair $I = 26$ (24–29; $n = 20$); pair II = 25 (23–26; $n = 20$); pair III = 31 (26–33; $n = 20$); pair IV = 31 (29–33; $n = 20$); pair $V = 27$ (25–30; $n = 20$); pair VI = 29 (27–31; $n = 20$); pair VII = 28 (26–30; $n = 20$). MCO comprising

articulated copulatory tube and accessory piece; total straight length 30 (27–33; $n = 20$). Copulatory tube a loose coil following sinuous path; 76 (75–77; $n = 13$) long. Accessory piece proximally enclosing base of copulatory tube to form frill-belted capsule; medial portion with three processes: primary process distally articulated to the capsule by lightly sclerotized ligament; secondary process grooved, closely associated with wedge-shaped tertiary process, serving as a guide for distal part of the tube; distal portion recurved, elongated, following medial part of the copulatory tube. Vagina a wavy tube, with enlarged funnel-shaped opening, 63 (57–70; $n = 20$) long.

**Molecular characterization:** The following DNA sequences were obtained for *D. ksibii*: partial 28S rDNA 792 bp long, partial 18S rDNA 478 bp long, ITS1 region 489 bp long, and 5.8S rDNA 11 bp long. Nine specimens of *D. ksibii* from three different river basins (Ksob, Oum Er'Rabia and Bouregreg) were sequenced (see Table 2 for accession numbers). Using molecular data, genetic variability was reported between specimens of *D. ksibii* parasitizing (i) *L. ksibi* collected from two different regions (Ksob River and Chbouka River), and (ii) different host species (*L. ksibi* and *L. rabatensis*). Pairwise distances calculated between *D. ksibii* and other *Dactylogyrus* species of Moroccan *Luciobarbus* are shown in Table 3.

**Remarks:** *El Gharbi, Birgi & Lambert (1994)* described *D. ksibii* from specimens collected on the gills of *L. ksibi* from the Ksob and Oum Er'Rabia Rivers, *L. magniatlantis* from the Tensift River (Tensift Basin) and *L. setivimensis* from the Boulahmayl (Bouregreg Basin) and Mellah (Mellah Basin) Rivers. However, host identification of the latter host species is probably erroneous (for the reason suggested above), with *L. rabatensis* apparently captured instead of *L. setivimensis* in the Boulahmayl River (Bouregreg Basin). All cyprinid specimens collected from the Bouregreg Basin during the present survey were identified as *L. rabatensis* (J. Vukic, pers. comm.). The Bouregreg Basin represents the current distribution area of this endemic Moroccan species (*Casal-López et al., 2015*). In the original description of *D. ksibii*, *El Gharbi, Birgi & Lambert (1994)* showed four iconotypes for the MCO, without indicating which corresponded to the specimens of *D. ksibii* found on the type host species and locality (i.e. *L. ksibi*, Ksob River). In addition, these authors reported morphometric variation in the haptoral sclerites among specimens of *D. ksibii* parasitizing different host species and occurring in different localities. *Rahmouni et al. (2017)* suggested that these morphological differences in the haptoral sclerites could indicate that *D. ksibii* represented a complex of several morphologically similar species. In the present study, specimens morphologically identified as *D. ksibii* were found on the gills of *L. ksibi* (=type host) and *L. rabatensis*. Subsequent DNA analysis revealed that the specimens of *D. ksibii* consisted of three genotypes, one found on *L. ksibi* collected in the Ksob River, one on the same host species but from the Chbouka River and one on *L. rabatensis* from the Grou River. A posteriori analysis of the specimens associated with the three genotypes revealed no morphological basis for splitting *D. ksibii* into two or three species (see Figs. 3A and 3B). Given that the level of morphological variation observed corresponds more to intraspecific variation, no new *Dactylogyrus* species are named at this time.

*Dactylogyrus ksibii* belongs to the group of congeners having a cross-shaped ventral bar (= 'carpathicus' type; *El Gharbi, Birgi & Lambert, 1994*). The MCO of *D. ksibii* most

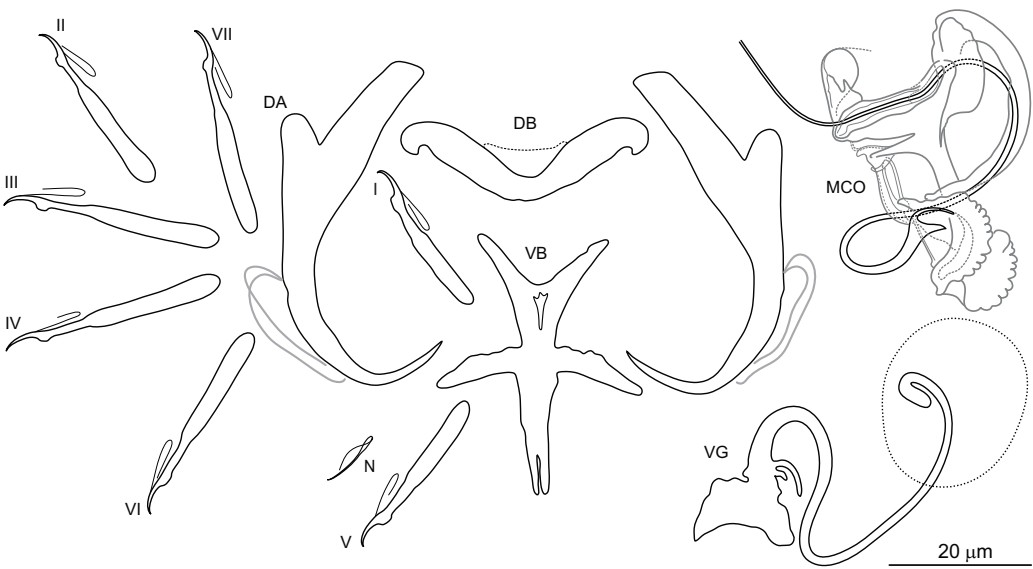

**Figure 4** **Sclerotized structures of *Dactylogyrus borjensis* ex *Luciobarbus zayanensis* (Oum Er'Rabia River, El Borj).** DA, dorsal anchor; DB, dorsal bar; VB, ventral bar; N, needle; I–VII, hooks (pairs I–V ventral; pairs VI, VII dorsal); MCO, male copulatory organ; VG, vagina.

closely resembles that of *D. scorpius* Rahmouni, Řehulková & Šimková, 2017, *D. benhoussai* Rahmouni, Řehulková & Šimková, 2017 and *D. varius* Rahmouni, Řehulková & Šimková, 2017, all three parasitizing species of Moroccan *Luciobarbus* and morphologically belonging to the 'scorpius' group (*Rahmouni et al., 2017*). *Dactylogyrus ksibii* differs from these three species by possessing a longer copulatory tube (76 vs 45 in *D. scorpius*; 76 *vs* 67 in *D. benhoussai*; 76 *vs* 65 in *D. varius*) and by details in the structure of the accessory piece. The accessory piece of *D. ksibii* is characterized by a medial part with a primary process closely associated with the secondary process (medial part with primary and secondary processes well defined/separated in *D. scorpius*, *D. benhoussai* and *D. varius*) and by a comparatively robust distal part (distal part smaller in *D. scorpius*, *D. benhoussai* and *D. varius*).

***Dactylogyrus borjensis*** El Gharbi, Birgi & Lambert, 1994 (Fig. 4)
**Type host:** *Luciobarbus zayanensis* Doadrio, Casal-López & Yahyaoui, 2016, previously referred to as *Luciobarbus nasus* [syn. *Barbus* (*Barbus*) *nasus*] (see Remarks).
**Type locality:** Oum Er'Rabia Basin, El Borj; Morocco.
**Other previously recorded hosts:** *Luciobarbus magniatlantis* (Pellegrin, 1919), previously referred to as *L. nasus* [syn. *B.* (*B.*) *nasus*] (*El Gharbi, Birgi & Lambert, 1994*) (see Remarks).
**Other previously recorded locality:** Tensift Basin, Zate River (Ait Ourir); Morocco (*El Gharbi, Birgi & Lambert, 1994*).
**Present host and localities:** *Luciobarbus zayanensis* Doadrio, Casal-López & Yahyaoui, 2016 (localities 3, 4).
**Site on host:** Gill lamellae.

**Specimens deposited:** Three vouchers (MNHN HEL1265-1267) and two hologenophores (MNHN HEL1268, 1269) from *L. zayanensis* (El Borj), and one voucher (IPCAS M-752) from *L. zayanensis* (Dar Oulad Zidouh).

**Description:** Based on 24 specimens fixed in GAP. Body length 624 (540–683; $n = 24$); greatest width 123 (100–130; $n = 24$) at level of ovary. One pair of anchors located dorsally: total length 47 (44–48; $n = 24$); length to notch 41 (40–42; $n = 24$); inner root 17 (15–19; $n = 24$) long; outer root 5 (4–6; $n = 24$) long; shaft curved, slightly swollen medially; point 12 (11–13; $n = 24$) long. One pair of needles located near hooks of pair V. Dorsal bar broadly V-shaped, with slightly rounded extremities, 36 (34–37; $n = 24$) long. Ventral bar cross-shaped, with five arms, 36 (34–38; $n = 24$) long, 28 (26–31; $n = 24$) wide. Seven pairs of hooks; each with delicate point, flattened thumb and shank inflated along proximal four-quarters; FH loop extending to level of shank inflation. Hook lengths: pair $I = 25$ (23–26; $n = 24$); pair II = 26 (25–28; $n = 24$); pair III = 31 (29–33; $n = 24$); pair IV = 31 (29–34; $n = 24$); pair V = 25 (23–27; $n = 24$); pair VI = 27 (26–30; $n = 24$); pair VII = 28 (26–30; $n = 24$). MCO complex, comprising articulated copulatory tube and accessory piece; total straight length 40 (36–42; $n = 24$). Copulatory tube a loose coil following sinuous path; 111 (107–120; $n = 12$) long. Accessory piece proximally enclosing base of copulatory tube to form frill-belted capsule; distal portion recurved, elongated, following medial part of copulatory tube; medial portion with three processes: primary process distally articulated to the capsule by lightly sclerotized ligaments; secondary process apically expanded into a wing like flap, closely associated with tertiary process serving as a guide for distal termination of the tube. Vagina a lightly sclerotized meandering tube, with disc-shaped opening, 92 (88–98; $n = 24$) long.

**Molecular characterization:** The following DNA sequences were obtained for *D. borjensis*: partial 28S rDNA 792 bp long, partial 18S rDNA 478 bp long, ITS1 region 489 bp long, and 5.8S rDNA 11 bp long. Three specimens of *D. borjensis* from the Oum Er'Rabia River were sequenced, and no intraspecific variation between the specimens of this species was noted. Pairwise distances between *D. borjensis* and the other *Dactylogyrus* species from Moroccan *Luciobarbus* are shown in Table 3.

**Remarks:** The type specimens of *D. borjensis* were not available (see *Rahmouni et al., 2017*). Based on the morphology of the haptoral and copulatory sclerites showed by *El Gharbi, Birgi & Lambert (1994)*, we consider our samples to be conspecific with this species. The original description is adequate except that these authors depicted the accessory piece of the MCO with the largest (secondary) medial process expanded into a tube through which the distal (recurved) part passes. As in other species of *Dactylogyrus* belonging to the "scorpius" group, the accessory piece is somewhat variable in this species, with the distal part bending inward to the right or left side of the medial part. This variability is likely a result of the compression used during preparation of individual worms for mounting. It appears likely that *El Gharbi, Birgi & Lambert (1994)* misinterpreted the distal part of the accessory piece as incorporated into the tubular part of the largest medial process, instead of lying under or above this part. In the present specimens of *D. borjensis*, the distal part of the accessory piece is not associated with any processes of the medial part. The largest process of the accessory piece is terminally extended to form wing like flap usually rolled

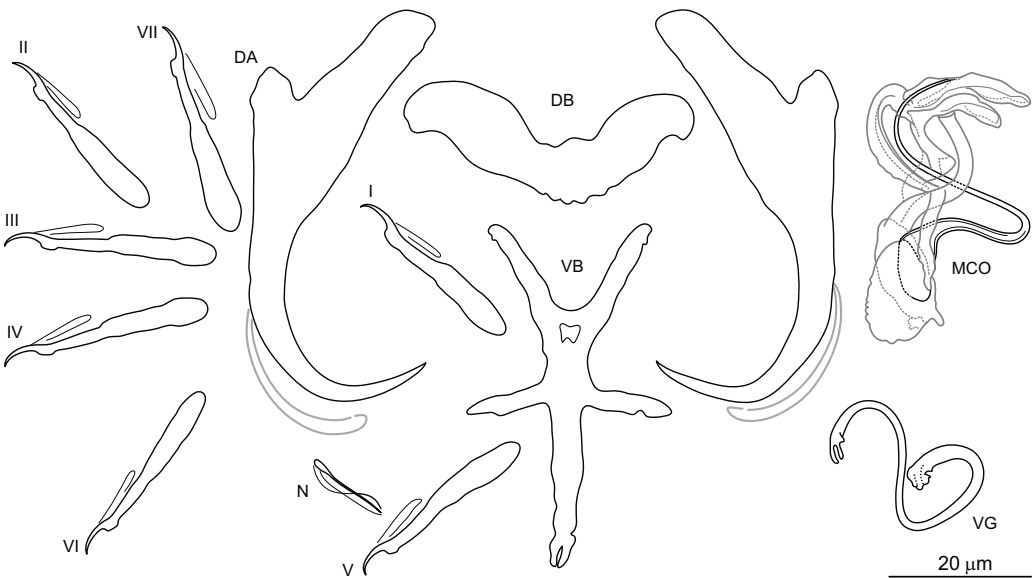

**Figure 5** Sclerotized structures of *Dactylogyrus draaensis* ex *Luciobarbus lepineyi* (Zouala Oasis). DA, dorsal anchor; DB, dorsal bar; VB, ventral bar; N, needle; I–VII, hooks (pairs I–V ventral; pairs VI, VII dorsal); MCO, male copulatory organ; VG, vagina.

in the shape of a sleeve, which serves as a guide for the distal part of the copulatory tube, and not for the distal part of the accessory piece as it is depicted in the original description of *D. borjensis*.

*Dactylogyrus borjensis* is most similar to *D. falsiphallus* Rahmouni, Řehulková & Šimková, 2017, described on the gills of *L. maghrebensis* from the Lahdar and Sebou rivers (Sebou Basin) (*Rahmouni et al., 2017*), in the general morphology of the haptoral sclerites and MCO. It differs from *D. falsiphallus* by having relatively robust and evenly sclerotized distal part of the accessory piece (distal part reduced in its sclerotization into a long spike in *D. falsiphallus*). In addition, the MCO is measurably smaller in *D. falsiphallus*.

*El Gharbi, Birgi & Lambert (1994)* listed *L. nasus* [syn. *B. (B.) nasus*] as the type host for this species. According to *Doadrio, Casal-López & Perea (2016)* the rheophilic *Luciobarbus* populations traditionally assigned to *L. nasus* (Günther, 1874) and *L. magniatlantis* (Pellegrin, 1919) comprise three species, each endemic for different basins in Morocco. *Luciobarbus nasus* is restricted to the Ksob Basin, *L. magniatlantis* to the Tensift Basin, and the recently described *L. zayanensis* to the Oum Er'Rabia Basin. Since the last basin represents the type locality for *D. borjensis*, we consider *L. zayanensis* as the type host of this species. The additional host record of *D. borjensis* on a population of *Luciobarbus* species previously referred to *L. nasus* from Tensift Basin (*El Gharbi, Birgi & Lambert, 1994*) should be attributed to *L. magniatlantis*.

**Dactylogyrus draaensis** El Gharbi, Birgi & Lambert, 1994 (Fig. 5)
**Type host:** *Luciobarbus pallaryi* (Pellegrin, 1919) [syn. *Barbus (Barbus) pallaryi*].
**Type locality:** Draa Basin, Dadès River, Ait Oudinar; Morocco.

**Other previously recorded locality:** Draa Basin, Tidili River, Iouine River; Morocco (*El Gharbi, Birgi & Lambert, 1994*).

**Present host and locality:** *Luciobarbus lepineyi* (Pellegrin, 1939) (locality 7).

**Site on host:** Gill lamellae.

**Specimens deposited:** Two vouchers (MNHN HEL1351, 1352) from *L. lepiney* (Zouala Oasis).

**Description:** Based on 30 specimens fixed in GAP. Body length 790 (645–901; $n = 16$); greatest width 120 (110–125; $n = 30$) at level of ovary. One pair of anchors located dorsally: total length 51 (48–55; $n = 30$); length to notch 39 (36–42; $n = 30$); inner root length 19 (17–21; $n = 30$); outer root length 5 (3–6; $n = 30$); point length 12 (11–14; $n = 30$). One pair of needles located near hooks of pair V. Dorsal bar broadly V-shaped, 41 (34–45; $n = 30$) long. Ventral bar cross-shaped, with five arms, 51 (37–56; $n = 30$) long, 36 (27–45; $n = 30$) wide. Seven pairs of hooks; each with delicate point, flattened thumb and shank inflated along proximal two-thirds; FH loop extending to level of shank inflation. Hook lengths: pair $I = 29$ (25–35; $n = 27$); pair II $= 29$ (25–32; $n = 29$); pair III $= 31$ (29–35; $n = 25$); pair IV $= 31$ (24–36; $n = 29$); pair $V = 28$ (24–33; $n = 29$); pair VI $= 31$ (27–38; $n = 28$); pair VII $= 31$ (28–36; $n = 25$). MCO complex, comprising articulated copulatory tube and accessory piece; total straight length 43 (40–52; $n = 30$). Copulatory tube sinuous, 116 (113–120; $n = 25$) long. Accessory piece proximally enclosing base of copulatory tube; medial portion with two auricle-like processes; distal portion recurved, following medial part of the copulatory tube. Vagina a lightly sclerotized meandering tube, 54 (47–60; $n = 30$) long.

**Molecular characterization:** The following DNA sequences were obtained for *D. draaensis*: partial 28S rDNA 792 bp long, partial 18S rDNA 478 bp long, ITS1 region 489 bp long, and 5.8S rDNA 11 bp long. Three specimens of *D. draaensis* from the Zouala Oasis were sequenced, with no intraspecific variability between specimens of this species noted. Pairwise distances between *D. draaensis* and the other *Dactylogyrus* species of Moroccan *Luciobarbus* are shown in Table 3.

**Remarks:** *Dactylogyrus draaensis* was originally described from the gills of *L. pallaryi* from two rivers in the Draa Basin (*El Gharbi, Birgi & Lambert, 1994*). The species is characterized by possessing a sinuous copulatory tube with the distal part supported by two auricle-like processes rising from the accessory piece. During the present survey, specimens provisionally identified as *D. draaensis* were collected from *L. lepineyi* in the Zouala Oasis. The drawings provided by *El Gharbi, Birgi & Lambert (1994)* were unclear regarding the morphology of the medial part of the accessory piece, hence we cannot state with certainty that our specimens are conspecific with those of *D. draaensis* from the Draa Basin. Assignment of a new species name to our specimens is not made at this time and will depend on re-collection of *D. draaensis* from the type host in the type locality for comparison with our specimens.

## Phylogenetic reconstruction

A concatenated sequence alignment (partial 28S, 18S rDNA and ITS1 combined) was used to construct a phylogenetic tree including 24 species, the alignment comprising 1458

aligned positions. Both ML and BI phylogenetic analyses yielded similar tree topologies. The BI tree is presented in Fig. 6, with bootstrap support values for ML and posterior probabilities for BI.

The phylogenetic reconstruction divided *Dactylogyrus* spp. parasitizing Moroccan cyprinids into three lineages. The first well-supported clade includes three species of *Dactylogyrus* (*D. kulindrii*, *D. volutus*, and *D. zatensis*) from Northwest African *C. fritschii* (Torinae) (Table 4). These species all have 'varicorhini' type of the anchors and bars: the anchors possess shaft turned into a point with a sharp-stepped narrowing from the inner side of the anchor; dorsal bar is saddle-shaped with posterior groove, usually giving the impression of butterfly wings; and ventral bar is V- or omega-shaped (*El Gharbi, Renaud & Lambert, 1992*; *Pugachev et al., 2009*). A sister relationship between *D. volutus* and *D. kulindrii* is also supported by morphological similarities in their MCOs, i.e., the copulatory tube is relatively wide in diameter and a simple accessory piece possesses two projections guiding the distal portion of the copulatory tube. The second clade, with several subclades, comprises species of *Dactylogyrus* collected from European and Moroccan species of *Barbus* and *Luciobarbus* (Table 4). *Dactylogyrus andalousiensis* El Gharbi, Renaud & Lambert, 1992 from Iberian *Luciobarbus sclateri* (Günther, 1868) (Portugal) was sister to a well-supported clade formed by nine species of *Dactylogyrus* parasitizing Moroccan *Luciobarbus*, all of which were characterized by (i) a cross-shaped ventral bar with five extremities (= 'carpathicus' or 'barbus' type; see *Pugachev et al., 2009*), (ii) an MCO possessing an accessory piece with the distal portion directed backwards along the circle of the curved copulatory tube (= 'chondrostomi' type; see *Pugachev et al., 2009*), a complex medial portion formed into ridge-shaped processes supporting the distal part of the copulatory tube and capsule-like proximal portion (= 'scorpius' subtype; *Rahmouni et al., 2017*). The basal position of *D. draaensis* in relation to the other *Dactylogyrus* spp. from Moroccan *Luciobarbus* was moderately supported by PP resulting from BI analysis and moderately/weakly supported by BP resulting from ML analysis. Four *Dactylogyrus* species formed the well-supported clade including two weakly supported assemblages: the first one represented by *D. benhoussai* and *D. varius* and the second one represented by *D. scorpius* and three genotypes of *D. ksibii*. All four *Dactylogyrus* species are morphologically very similar, both in the haptoral structures and the MCOs. Finally, *D. marocanus* formed a separated lineage supported by differing morphology.

## DISCUSSION

Since the anatomical details of dactylogyrids are generally poorly known, discrimination between species of *Dactylogyrus* relies chiefly on the morphometric characteristics of sclerotized structures of the haptor and reproductive organs. In recent years, however, molecular phylogenetic analysis has revealed hidden genetic variation and/or cryptic species within this genus. *Rahmouni et al. (2017)* reported two cryptic species of *Dactylogyrus* (*D. benhoussai* and *D. varius*) on two species of Moroccan *Luciobarbus* (*L. yahyaouii* and *L. maghrebensis*, respectively), while *Benovics et al. (2018)* revealed potential cryptic species complexes within three species of *Dactylogyrus* (*Dactylogyrus rutili* Gläser, 1965,

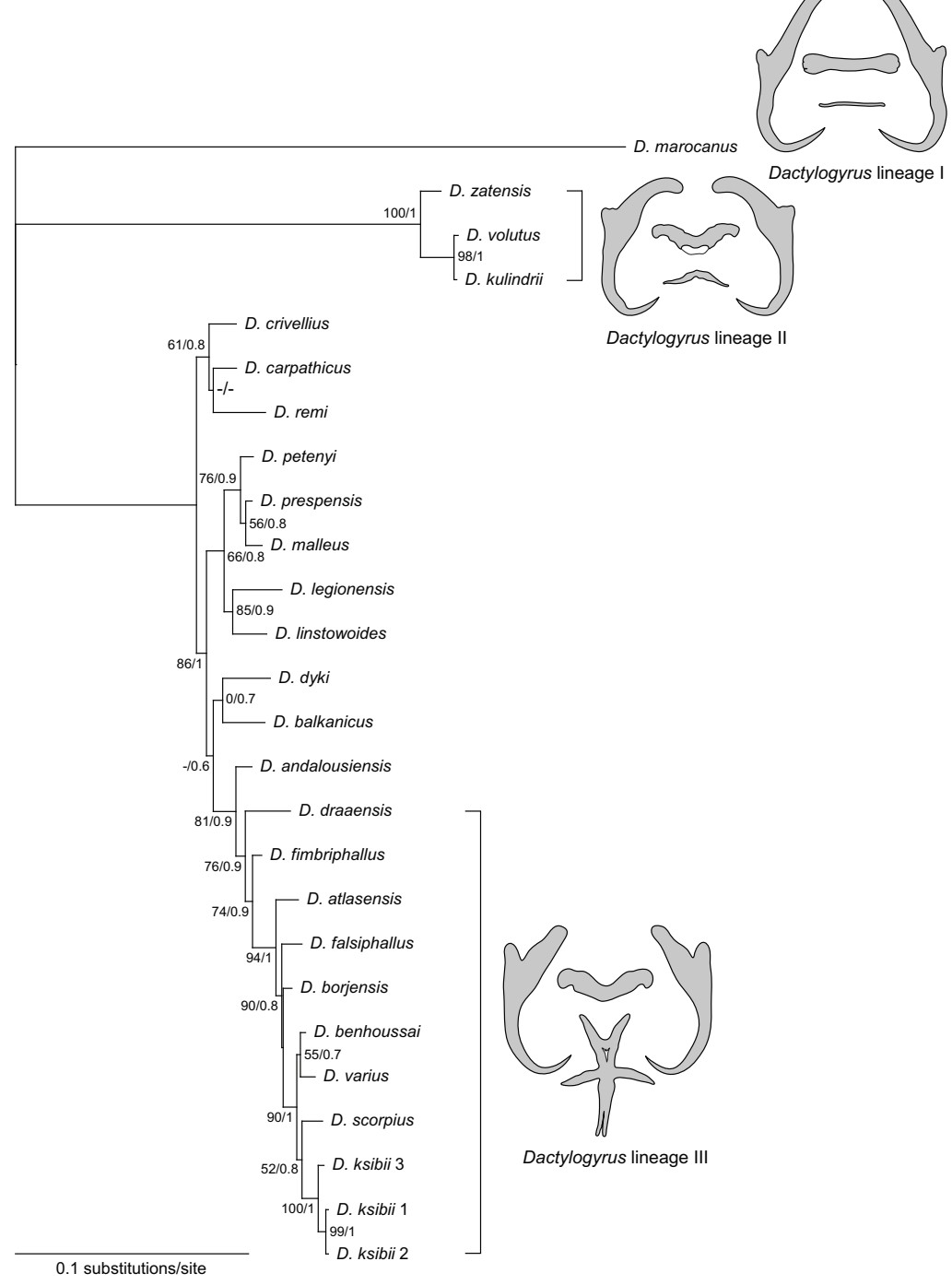

**Figure 6** **Phylogenetic tree constructed by BI analysis using combined data of partial 28S, 18S rDNA and ITS1.** Values showed at the nodes indicate posterior probabilities from BI analysis (only values higher than 0.7) and bootstrap values from ML analysis (only values higher than 50). *Dactylogyrus* species parasitizing Moroccan cyprinids form three lineages, each with characteristic type of haptoral configuration: 'pseudanchoratus' type (lineage I; *D. marocanus*), 'varicorhini' type (lineage II; *D. zatensis*), and 'carpathicus' type (lineage III; *D. scorpius*).

**Table 4 List of *Dactylogyrus* species, their cyprinid host species, country of collection and GenBank accession numbers for sequences used for the phylogenetic reconstruction.**

| *Dactylogyrus* species | Host species | Locality | 28S rDNA | 18S rDNA ITS1 |
|---|---|---|---|---|
| *D. andalousiensis* El Gharbi, Renaud & Lambert, 1992 | *Luciobarbus sclateri* (Günther) | Portugal | KY629351 | KY629331 |
| *D. atlasensis* El Gharbi, Birgi & Lambert, 1994 | *Luciobarbus pallaryi* (Pellegrin) | Morocco (Zouala Oasis) | KY629356 | KY629337 |
| *D. balkanicus* Dupont & Lambert, 1986 | *Barbus prespensis* Karaman | Albania | KY201107 | KY201093 |
| *D. benhoussai* Rahmouni et al., 2017 | *Luciobarbus yahyaouii* Doadrio, Casal-López & Perea | Morocco (Melloulou River) | KY553862 | KY578025 |
| *D. borjensis* El Gharbi, Birgi & Lambert, 1994 | *Luciobarbus zayanensis* Doadrio, Casal- López & Yahyaoui | Morocoo (Oum Er'Rabia River) | MN973819 | MN974257 |
| *D. carpathicus* Zachvatkin, 1951 | *Barbus barbus* (Linnaeus) | Czech Republic | KY201111 | KY201098 |
| *D. crivellius* Dupont & Lambert, 1986 | *Barbus peloponnesius* Valenciennes | Greece | KY201108 | KY629339 |
| *D. draaensis* El Gharbi, Birgi & Lambert, 1994 | *Luciobarbus lepineyi* (Pellegrin) | Morocco (Zouala Oasis) | MN973816 | MN974258 |
| *D. dyki* Ergens & Lucký, 1959 | *Barbus barbus* | Czech Republic | KY629367 | KY629338 |
| *D. falsiphallus* Rahmouni et al., 2017 | *Luciobarbus maghrebensis* Doadrio, Perea & Yahyaoui | Morocco (Lahdar River) | KZ553861 | KY57802 |
| *D. fimbriphallus* El Gharbi, Birgi & Lambert, 1994 | *Luciobarbus massaensis* (Pellegrin) | Morocco (Massa River) | KY629357 | KY629332 |
| *D. ksibii* 1 El Gharbi, Birgi & Lambert, 1994 | *Luciobarbus ksibi* (Boulenger) | Morocco (Ksob River) | MN973812 | MN974252 |
| *D. ksibii* 2 | *Luciobarbus ksibi* | Morocco (Chbouka River) | MN973811 | MN974251 |
| *D. ksibii* 3 | *Luciobarbus rabatensis* Doadrio, Perea & Yahyaoui | Morocco (Grou River) | MN973817 | MN974250 |
| *D. kulindrii* El Gharbi, Birgi & Lambert, 1994 | *Carasobarbus fritschii* (Günther) | Morocco (Ksob River) | KY629354 | KY629336 |
| *D. legionensis* González -Lanza & Alvarez-Pellitero, 1982 | *Luciobarbus guiraonis* (Steindachner) | Spain | KY629350 | KY629330 |
| *D. linstowoides* El Gharbi, Renaud & Lambert, 1992 | *Luciobarbus guiraonis* | Spain | KY629349 | KY629329 |
| *D. malleus* Linstow, 1877 | *Barbus barbus* | Czech Republic | KY201112 | KY201099 |
| *D. marocanus* El Gharbi, Birgi & Lambert, 1994 | *Carasobarbus fritschii* | Morocco (Oum Er'Rabia River) | KY629355 | KY629333 |
| *D. petenyi* Kaštak, 1957 | *Barbus balcanicus* Kotlík | Greece | KY201113 | KY201097 |
| *D. prespensis* Dupont & Lambert, 1986 | *Barbus prespensis* | Greece | KY201110 | KY201096 |
| *D. remi* Řehulková, Benovics & Šimková, 2020 | *Luciobarbus graecus* (Steindachner) | Greece | KY201115 | KY201101 |
| *D. scorpius* Rahmouni et al., 2017 | *Luciobarbus rifensis* Doadrio, Casal- López & Yahyaoui | Morocco (Loukkos River) | KY553860 | KY578023 |
| *D. varius* Rahmouni et al., 2017 | *Luciobarbus maghrebensis* Doadrio, Perea & Yahyaoui | Morocco (Lahdar River) | KZ553863 | KY578026 |
| *D. volutus* El Gharbi, Birgi & Lambert, 1994 | *Carasobarbus fritschii* | Morocco (Lahdar River) | KY629353 | KY629334 |
| *D. zatensis* El Gharbi, Birgi & Lambert, 1994 | *Carasobarbus fritschii* | Morocco (Oum Er'Rabia River) | KY629352 | KY629335 |

*Dactylogyrus dyki* Ergens & Lucký, 1959 and *Dactylogyrus ergensi* Molnár, 1964) parasitizing Balkan cyprinids. As different ribosomal DNA molecular markers show differing rates of evolution, they are appropriate for evaluating genetic divergence at different levels (*Huelsenbeck & Ronquist, 2001*). In the present study, we applied genetic markers widely used for monogeneans, i.e., 18S rDNA, 28S rDNA and the ITS1 region (*Cunningham, 1997*; *Meinilä et al., 2002*; *Ziętara & Lumme, 2002*). Molecular characterization of *Dactylogyrus* species in our study showed that specimens identified morphologically as *D. ksibii*, collected from the gills of two different hosts (*L. ksibi* and *L. rabatensis*) and geographically distant basins, exhibited a low level of intraspecific variability (0.8% for the combined 18S rDNA and ITS1 sequences). Molecular divergence observed between specimens of *D. ksibii* (i.e., those parasitizing *L. ksibi* from Ksob River, *L. ksibi* from Chbouka River, and *L. rabatensis* from Grou River) may be explained by the large geographical distances between parasites and hosts precluding gene flow between isolated populations. Our findings indicate that specimens of *D. ksibii* reported by *El Gharbi, Birgi & Lambert (1994)* from *Luciobarbus* species inhabiting various river basins in Morocco need serious reinvestigation in the future using morphological and molecular data.

Besides providing crucial taxonomic information on species, the sclerotized structures of the haptor and reproductive organs are of particular interest in evolutionary studies focused on the link between morphological and molecular interspecific similarities of *Dactylogyrus* spp. (e.g., *Benovics, Kičinjaová & Šimková, 2017*; *Benovics et al., 2018*). In addition, as the morphology of the attachment organ is usually viewed as the result of adaptive processes to the host microenvironment (e.g., *Kearn, 1994*; *Vignon & Sasal, 2010*), the morphological characteristics of the haptor may have the potential to reflect the phylogeny and historical biogeographical routes of their hosts (e.g., *Šimková et al., 2006*). On the basis of morphological resemblance, many *Dactylogyrus* species can be grouped into morphological types, often derived from a single structure (*Pugachev et al., 2009*). Some of these groups may be considered as phylogenetic units, members of one unit often parasitizing closely related hosts. However, it is relatively difficult to determine to what extent morphological features reflect a phylogenetic signal unless adaptive forces associated with the possibility of host switching can be excluded.

Based on the shape of the ventral bar, species of *Dactylogyrus* parasitizing *Luciobarbus* spp. are grouped into four morphological types, i.e., those with a rod-shaped, omega-shaped, inverted T-shaped or cross-shaped ventral bar (see *Pugachev et al., 2009*). All Moroccan species of *Dactylogyrus* that are host-specific to *Luciobarbus* spp. belong to the group with a cross-shaped ventral bar, where the anterior arm is widely bifurcated and the posterior arm is more or less split (= five radial type; 'carpathicus' or 'barbus' type in *El Gharbi, Birgi & Lambert (1994)* and *Pugachev et al. (2009)*, respectively). *Dactylogyrus* spp. with this type of ventral bar have also been recorded on *Luciobarbus* spp. inhabiting the Balkan Peninsula (*Řehulková, Benovics & Šimková, 2020*) and the region around the Caspian Sea (*Pugachev et al., 2009*), and on *Aulopyge huegelii* (*Benovics, Kičinjaová & Šimková, 2017*) in the Balkans (*Benovics, Kičinjaová & Šimková, 2017*). The majority of *Dactylogyrus* species (except *D. andalousiensis* and *Dactylogyrus linstowoides*) reported on *Luciobarbus* spp. on the Iberian Peninsula have V- or omega-shaped ventral bars. Inasmuch as the haptor

has to be evolutionary adapted as far as possible to the host microenvironment (e.g., *Kearn, 1968*; *Šimková et al., 2001*), the same morphological type of haptor in Moroccan and Balkan-Caspian *Dactylogyrus* spp. may suggest that *Luciobarbus* spp. inhabiting these regions share a common ancestor. This assumption is also supported by *Tsigenopoulos et al. (2003)* and *Yang et al. (2015)*, who showed that most *Luciobarbus* spp. from Northwest Africa are more closely related to *Luciobarbus* spp., from the Middle East than to those from the Iberian Peninsula. In addition, on the basis of phylogenetic reconstruction, *Šimková et al. (2017)* demonstrated that species of *Dactylogyrus* parasitizing northwest African *Luciobarus* have a European/west Asian origin. When looking at MCO morphology, the 'Moroccan' and 'Balkan-Caspian' species of *Dactylogyrus* with a cross-shaped ventral bar are characterized by different types of MCO, i.e., the 'chondrostomi' and 'kulwieci' types, respectively (*Pugachev et al., 2009*). However, the 'chondrostomi' group, characterized by an accessory piece with the distal portion directed backwards along the circle of the curved copulatory tube, is one of the most specious groups within the Palaearctic species of *Dactylogyrus* exhibiting different types of ventral bar (*Pugachev et al., 2009*). On the basis of detailed morphology of the accessory piece, it would appear that Moroccan species of *Dactylogyrus* form another subgroup within the 'chondrostomi' group, named by *Rahmouni et al. (2017)* as the 'scorpius' subgroup. This subgroup is characterized by an accessory piece with a complex medial portion formed into ridge-shaped processes supporting the distal part of the copulatory tube and a relatively massive capsule-like proximal portion. It would be interesting to further investigate the degree of relatedness between Moroccan and other *Dactylogyrus* spp. with the 'chondrostomi' type of MCO, with the intention of answering the question of whether the similarity between MCOs is a result of homoplasy or shows a phylogenetic signal.

Concerning host specificity, *D. marocanus* exhibits an unusually broad host range that includes phylogenetically distant host species. Our records of *D. marocanus* on the gills of *L. zayanensis* and *P. maroccana* increase the range of available host species such that the host range now includes eight species of four cyprinid genera representing two phylogenetic lineages, i.e., the Torinae, including *Carasobarbus*, *Labeobarbus* and *Pterocapoeta*, and the Barbinae, including *Luciobarbus*. *El Gharbi, Birgi & Lambert (1994)* suggested that *C. fritschii* [syn. *Labeobarbus fritschii*] represents the original host of *D. marocanus*; however, *Šimková et al. (2017)* showed that *D. marocanus* is closely related to *Dactylogyrus* spp. parasitizing West African species assigned to *Labeo*, which suggests a host-switch from African labeonins to *C. fritschii* and the other reported northwest African cyprinid hosts. In terms of morphology, *D. marocanus* is the only Moroccan *Dactylogyrus* species belonging to the 'pseudanchoratus' group (*El Gharbi, Birgi & Lambert, 1994*), which includes *Dactylogyrus* spp. reported on cyprinids (mostly species of *Labeo*) from the wider equatorial region in Africa (*Paperna, 1979*) and, interestingly, *Dactylogyrus* spp. parasitizing *Garra rufa* (Labeoninae) in Iran (*Gussev, Jalali & Molnár, 1993*; *Pugachev et al., 2009*). The molecular phylogeny of labeonins provided by *Tang, Getahun & Liu (2009)* showed that the Asian *G. rufa* and African *Garra* spp. formed sister groups, and that this Afro-Asian clade was nested within a larger clade containing all other Asian *Garra* spp. In addition, their study supports an East Asian origin of labeonins, and into-Africa dispersal

events for the African species of *Garra* and *Labeo*. If the phylogeny of highly host-specific parasites follows the phylogeny and historical biogeography of their hosts, it would be interesting to analyze the phylogenetic relationship between *Dactylogyrus* spp. parasitizing *G. rufa* and those of the 'pseudanchoratus' group (including *D. marocanus*) parasitizing African labeonins. The close relationship between these *Dactylogyrus* spp., supported by both morphological and molecular data, could point to their common Asian origin.

In accordance with the study of *Šimková et al. (2017)* we showed that *Dactylogyrus* species parasitizing northwest African cyprinid fishes belong to three evolutionary lineages. *Dactylogyrus* spp. from Moroccan species of *Luciobarbus* (Barbinae) represent the largest (monophyletic) group in number of species. High similarity in morphology of the sclerotized structures in these parasites, together with high host specificity, could suggest rapid diversification following the geographical separation and diversification of *Luciobarbus* spp. The basal position of *D. andalousiensis* parasitizing Iberian *L. sclateri* in relation to the monophyletic group including *Dactylogyrus* spp. parasitizing Moroccan *Luciobarbus* spp. (Barbinae) strongly supports a European origin for this group of parasites, as previously shown by *Šimková et al. (2017)*. *Dactylogyrus* spp. parasitizing fishes of Moroccan Torinae (species of *Carasobarbus* and *Labeobarbus*) form the second largest group characterized by the 'varicorhini' morphological type of sclerotized structures (*El Gharbi, Birgi & Lambert, 1994*), probably originating from Asian cyprinids (*Šimková et al., 2017*). Mapping the evolutionary history of *D. marocanus* is difficult due to its low level of host specificity; however, morphological and molecular data suggest an affinity of this species to *Dactylogyrus* spp. parasitizing African labeonins (*Šimková et al., 2017*). The phylogenetic position of *Dactylogyrus guirensis* El Gharbi, Birgi & Lambert, 1994, the only Moroccan species of the 'guirensis' morphological type (*El Gharbi, Birgi & Lambert, 1994*), remains unresolved as no molecular data are available for this species at this time.

## CONCLUSION

The present study represents one more step towards the understanding of morphological and phylogenetic relationships among species of *Dactylogyrus* (i.e., at the levels of inter- and intraspecific variability) parasitizing cyprinoid fishes. Our phylogenetic reconstruction of *Dactylogyrus* species from Moroccan cyprinids supports that the morphological characters formerly proposed by *El Gharbi, Birgi & Lambert (1994)* and further developed by *Rahmouni et al. (2017)* to group species of *Dactylogyrus* are relevant. Our results also show that a detailed morphological analysis of closely related monogeneans is an essential part of integrative taxonomy, not only for accurate delimitation of species (or justifying of cryptic species) but also for searching the link between morphology and molecules in phylogenetic and cophylogenetic studies. This is especially true for species of *Dactylogyrus* undergoing probable rapid speciation without discernible morphological differentiation, such as some of those parasitizing Moroccan cyprinids.

## ACKNOWLEDGEMENTS

We are grateful to Jasna Vukić from Faculty of Science, Charles University, Prague, for fish collection, determination and valuable discussion concerning current taxonomy of *Luciobarbus* spp. Thanks are due to Mária Lujza Červenka Kičinja and Tomáš Pakosta for their help with the fish examination, parasite collection and fixation. We are also grateful to Delane Kritsky and two anonymous reviewers whose comments greatly improved the manuscript. We thank Professor Abdelaziz Benhoussa from Mohamed V University of Rabat for the help in the fish collection. We kindly thank Kevin Roche for the English revision of the final draft.

### Funding

This study was supported by the Czech Science Foundation (project no. 15-19382S). The funders had no role in study design, data collection and analysis, decision to publish, or preparation of the manuscript.

### Grant Disclosures

The following grant information was disclosed by the authors:
Czech Science Foundation: 15-19382S.

### Competing Interests

The authors declare there are no competing interests.

### Author Contributions

- Eva Řehulková conceived and designed the experiments, performed the experiments, analyzed the data, prepared figures and/or tables, authored or reviewed drafts of the paper, and approved the final draft.
- Imane Rahmouni performed the experiments, analyzed the data, prepared figures and/or tables, authored or reviewed drafts of the paper, and approved the final draft.
- Antoine Pariselle performed the experiments, authored or reviewed drafts of the paper, and approved the final draft.
- Andrea Šimková conceived and designed the experiments, analyzed the data, authored or reviewed drafts of the paper, and approved the final draft.

### Animal Ethics

The following information was supplied relating to ethical approvals (i.e., approving body and any reference numbers):

All applicable institutional, national, and international guidelines for the care and use of animals were followed. This study was approved by the Ethics Comittee of Masaryk University (Czech Republic) (approval number CZ01302).

## Field Study Permissions

The following information was supplied relating to field study approvals (i.e., approving body and any reference numbers):

Field experiments were approved by the Haut Commissairiat aux Eaux et Forêts et à la Lutte contre la Désertification (Ministère de l'Agriculture, de la Pêche Maritime, du Développement Rural et des Eaux et Forêts, Royaume du Maroc) (N°62 HCEFLCD/DLCDPN/CPC/PPC).

## DNA Deposition

The following information was supplied regarding the deposition of DNA sequences:

*D. marocanus* sequences are available at GenBank: MW218580, MW218673, MW218669, MW218671, MW218579, and MW218672.

## Data Availability

The raw measurements of all *Dactylogyrus* spp. are available as a Supplemental File.

## Supplemental Information

Supplemental information for this article can be found online at http://dx.doi.org/10.7717/peerj.10867#supplemental-information.

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
