# Peer review of "Integrating morphological and molecular approaches for characterizing four species of Dactylogyrus (Monogenea: Dactylogyridae) from Moroccan cyprinids, with comments on their host specificity and phylogenetic relationships"

_PeerJ, doi:10.7717/peerj.10867_

## Round 0.1 · original submission · Minor Revisions

I apologize for the delay. One of the reviewers communicated with the journal staff and obtained a long delay for sending the review.

I am also attaching a document sent to me by reviewer #2 by email.

Finally we have three reviews, which each suggest only minor changes. However, the addition of all these minor suggestions makes, finally, a number of things to change.

Reviewer 1 ·

Basic reporting

This is very interesting and important papers that integrates morphological and molecular approaches for characterizing species of Dactylogyrus from Moroccan Luciobarbus. All experiments are correct, statistical analysis is sound and conclusions are well supported. Literature is well referenced and relevant, general structure conforms to PeerJ standards, figures are relevant, high quality, well labelled and described. The manuscript is clearly written in professional, unambiguous language. Having said that, I would like to note that the manuscript has a number of weaknesses which should be addressed.

The M&M section includes the well-described subsection entitled as “DNA isolation, amplification and sequencing”, but the results of this work is poorly present in the manuscript. I mean that authors should provide the table with information on host species, locality and accession number (or place where it will be inputted if the manuscript will be accepted) for each isolate obtained in the present study. Moreover, the phylogenetic analysis presented in the manuscript completely includes sequences from already published papers or papers in press.

Subsection “Molecular characterization” provided to each species of studied Dactylogyrus contains information with the number of specimens that were sequenced and short information about intraspecific genetic variability, but again I did not see any information and analytic data, e.g. isolates deposited, counted pairwise genetic distances, phylogenetic tree.
L197, 258-260, 346, 421 Numbers for hologenophores deposited in the museum are provided, but information on the number of corresponding isolates deposited in GenBank is absent
Concluding the above authors should:
1) Deposit sequenced isolates of species in GenBank
2) Provide additional table with information on such isolates (at least host and locality data, accession numbers).
3) Provide an accession number of the sequenced isolate to each hologenophore
4) Provide graphical information (table and/or tree) that would reflect the results of molecular analysis for isolates sequenced in the present study (see also comment below).

The discussion section does not include the conclusion paragraph with a general conclusion and future perspectives of the study.

Minor comments
L23-26 Four species of studied Dactylogyrus spp. parasitizing on Luciobarbus characterized as those, which have “an attachment organ with a cross-shaped (five radial) ventral bar and the male copulatory organ containing an accessory piece with a distal portion directed backwards along the circle of the curved copulatory tube”. But Dactylogyrus marocanus does not fit to the provided description. The abstract should be corrected.

L218-220. Authors stated that “Seven specimens from three host species (C. fritschii, L. ksibi, and P. maroccana) were sequenced, with no intraspecific genetic variability between specimens parasitizing different host species found.” I do not believe that seven specimens of Dactylogyrus marocanus from three host species were identical. Please provide a table with counted pairwise genetic distances between them. You can allocate this table in Supplementary data.

L462. In the sentence “Both phylogenetic analyses yielded similar tree topologies” Insert “ML and BI” between “Both” and “phylogenetic”
Check spelling Dactylogyrus marocanus through the text and tables. There is also D. maroccanus. See L 170 and Table 3
Figure 6. Please provide more detailed information about tree, namely used genes, tree length.
Authors should explain while Dactylogyrus marocanus is out of phylogenetic tree.

Experimental design

no comment

Validity of the findings

no comment

Additional comments

no comment

·

Basic reporting

I am recommending acceptance after minor revision and consideration of the recommendations and remarks which I have placed directly on the manuscript. The English is very good. The following are the most important concerns:
1) higher taxa are not host or parasites, although their member species might be.
2) My major concern is the apparent arbitrary placement of the root in the phylogenetic tree. As it is, there is no evidence that its placement is correct in the tree. By including arbitrary root (which clearly is a result of the authors' bias), the results are misleading and more than likely erroneous. Remove the root from the tree and let your readers deal with the fact that on is not provided, which by the way, can only be done in a phylogenetic analysis by including an outgroup taxon (which you haven't done). You might also want to reconsider what you have said in the discussion re the tree as well.
3) The mixing of classifications (see ms.)
4) Soft non-repeatable measurements (see ms.)

Experimental design

ok

Validity of the findings

ok

Reviewer 3 ·

Basic reporting

The paper is well-written. The introduction provides a clear overview of the state-of-the-art, and situates the current research very well in what has already been done. The use of the English language is, as far as I can tell, fine.

l. 52: do you mean recent descriptions of fish or of parasite species?
l. 383: "specimens" + "was": confusion of singular and plural?
l. 435-437: why no elaborate description of the morphology of the MCO in Dactylogyrus draaensis?
l. 525-527: "unless [...] cannot be excluded": something wrong in this sentence structure
L. 564-565: "Our records [...] and increases...": something wrong in this sentence structure
l. 577: "Getahun", not "Getahum"
Fig. 3A: this species was collected from this host species on two different localities; from which locality did the specimen(s) on which this drawing was based, come?
Fig. 6: perhaps it would be nice to guide the reader through the tree by indicating the clades mentioned in the text, or the host species, or by highlighting the Moroccan species...

General question: is it the first time sequences of the species in question are published? Perhaps it can also be clearly indicated in Table 2 which sequences are new in this study.

Experimental design

The methods are well-explained, and correctly used. I only have some questions:

l. 143: why did the authors opt for the use of uncorrected p-distances if they later select a model of molecular evolution?
l. 145-148: please explain why these species were chosen to be included in the phylogenetic analyses. On the basis of which criteria were they selected? For example, does the analyses include all Moroccan representatives of Dactylogyrus for which sequences are available?
l. 156-158: which partition was used for the concatenated dataset, and how were the selected models implemented? Not all of them are available in the phylogenetic packages used.

Validity of the findings

l. 144 suggests that the taxonomic treatments of the species under study are redescriptions. However, if that is the case, it would be useful to stress whether the authors report any new phenotypical aspects, or any differences in comparison to the original species descriptions. (The authors did this nicely for Dactylogyrus borjensis.)
l. 240-242: did the authors check for size differences between the parasites of different hosts?
l. 275-276: isn't this also the case in D. marocanus and D. draaensis? On Figs. 2 and 5, I also see this expansion of the proximal part (although to a lesser extent than in D. ksibii perhaps), but for D. marocanus and D. draaensis this aspect is not mentioned in the text. Perhaps I don't see what the authors intended to show here of course, but in that case the difference should be made clearer in the drawings.
l. 485: A PP of 0.9 is not "well supported".
l. 486: I do not see the bootstrap value supporting the relationship of D. borjensis towards this clade.
l. 587-588: this tree in itself says nothing about independent origins, as the sister lineages of two of the three clades to which Moroccan representatives belong, are not represented.
Hence, perhaps the conclusions based on the tree should be slightly nuanced.

Additional comments

The phylogenetic coverage of this research project in its entirety (also beyond this manuscript: summarised in the introduction) is impressive in general, and the present manuscript provides a nice overview of what is known on the species in question. I recommend its publication, pending some improvements and clarifications as suggested above.

---

## Round 0.2 · Minor Revisions

One of the three reviewers of the first version did not respond and thus you receive now two reviews only. One reviewer (Delane) considers that the paper can be published as it, but the other (anonymous) suggests a few more improvements. I am confident that this will not cause any problem and I will be happy to accept the revised version without sending it again to the reviewers.

·

Basic reporting

I've looked at the authors' responses, and while they have ignored several, I don't see where a new response to their decisions would make any difference. So go ahead with publication and let the authors' be responsible for their decisions.

Experimental design

see above

Validity of the findings

see above

Additional comments

see above

Reviewer 3 ·

Basic reporting

L. 577-579: I agree the sentence is grammatically correct, but its meaning is not… I think you mean that it is hard to say whether worms with a similar morphology are also closely related, because it could be that they share a similar morphology because of adaptations to a similar host, rather than because of phylogenetic relationships. Correct? And therefore the only way to be sure that morphology mirrors phylogeny, is to exclude such adaptive phenomena in the morphology. If that is what you mean, the sentence should read “… CAN be excluded”.


Fig. 6: the caption should explain what the scale bar represents.

Experimental design

no comment

Validity of the findings

L. 537-538: in the previous version I had a question about the statistical support for the sister-group relationship between Dactylogyrus borjensis and the clade consisting of D. benhoussai + D. varius + D. ksibii + D. scorpius. The authors’ reply gives me the impression that this node is supported by a Bayesian PP below 70% and a ML bootstrap below 50%. If this is the case, this node should perhaps best be collapsed, but more importantly: it cannot be concluded whether D. borjensis or rather D. falsiphallus are sister to this well-supported clade of D. benhoussai + D. varius + D. ksibii + D. scorpius.

Additional comments

I thank the authors for the constructive discussion. They have nicely implemented the suggestions of the referees, as far as I can see; they have carefully formulated and justified their responses, and I agree with their arguments in almost all cases. There are two relatively minor elements where our opinions seem to differ, and a single small addition to a figure caption that I believe is needed. If these three things can be addressed (the editor can check this as far as I’m concerned), this manuscript is absolutely ready for publication.

Well done, and my best wishes for a safe 2021 for the authors!

---

## Round 0.3 · accepted · Accept

This is a nice paper, a good addition to the taxonomy of Dactylogyrus.